# A method for intuitively extracting macromolecular dynamics from structural disorder

Nicholas M. Pearce [1,2 ✉] & Piet Gros [1]

Macromolecular dynamics manifest as disorder in structure determination, which is subsequently accounted for by displacement parameters (also called temperature factors, or B-factors) or alternate conformations. Though B-factors contain detailed information about structural dynamics, they are the total of multiple sources of disorder, making them difficult to interpret and thus little-used in structural analysis. We report here an analytical approach for decomposing molecular disorder into a parsimonious hierarchical series of contributions, providing an intuitive basis for quantitative structural-dynamics analysis. We demonstrate the decomposition of disorder on example SARS-CoV-2 and STEAP4 structures, from both crystallographic and cryo-electron microscopy data, and reveal how understanding of the macromolecular disorder leads to deeper understanding of molecular motions and flexibility, and suggests hypotheses for molecular mechanisms.

[1] Structural Biochemistry, Bijvoet Centre for Biomolecular Research, Department of Chemistry, Faculty of Science, Utrecht University, Utrecht, The Netherlands. [2] Department of Chemistry and Pharmaceutical Sciences, VU Amsterdam, Amsterdam, The Netherlands. ✉email: n.m.pearce@vu.nl

Macromolecular crystallography records a temporal and spatial average over billions of copies of a molecule in a crystal, and small variations in the relative positions and orientations of (parts of) these molecules lead to a blurring of the observed electron density[1]. There are many potential sources of disorder within a crystal (Fig. 1): crystal imperfections lead to global disorder contributions for the whole unit cell[2]; static or dynamic molecular displacements leads to systematic disorder over whole molecules, or sections of molecules[3]; and finally, atomic disorder captures the individual dynamics of an atom relative to its surroundings. The same is true for cryo-electron microscopy data: while large-scale structural changes lead to images being separated into distinct classes, global errors in image alignment and local structural differences lead to disorder equivalent to that of crystallography. Disorder is thus a fundamental feature of macromolecular structural data, and explicitly encodes information about local conformational dynamics. Conversely, disorder is a relatively untapped source of structural information: large-scale disorder obscures local disorder[4], thereby limiting any biological interpretations.

Local continuous disorder in a macromolecular atomic model is described using atomic displacement parameters (ADPs; also called temperature factors, or B-factors), which describe gaussian distributions of atomic positions. At high resolution, anisotropic ADPs (a-ADPs) can be used[5]; at more moderate resolutions, we use isotropic ADPs (i-ADPs), often in combination with Translation-Libration-Screw (TLS) models that generate sets of a-ADPs describing collective rigid-body motions for groups of atoms[3,6,7]. For lower resolutions, one i-ADP per residue or region are even possible. We note that in other works, ADPs may be used to mean anisotropic displacement parameters instead of atomic displacement parameters; to allow unambiguous usage, we use ADPs for atomic displacement parameters, and a-ADPs or i-ADPs to refer to anisotropic ADPs or isotropic ADPs, respectively.

When B-factors are displayed, they are shown as spheres (for i-ADPs) or ellipsoids (for a-ADPs, also sometimes called thermal ellipsoids). The size of a sphere or ellipsoid in a particular direction is indicative of the disorder in that direction. More precisely, the surface of the sphere or ellipsoid is a probability contour, which contains the atom position within the surface a chosen fraction of the time.

Comparative analysis of disorder between macromolecular structures is dominated by the empirical B-factor normalisation approach[8], which allows only for qualitative, relative comparisons of i-ADPs[4]. Additionally, while it is possible to extract physical motions from refined TLS descriptions[9,10], such analyses may conflate multiple sources of disorder (e.g., analysis of domain motions will also contain contributions from molecular disorder), and may not, in any case, result in physical motions[9]. On the other hand, at high resolution, where models are most likely to contain atomic-level dynamical information, there are no general methods for deconstructing a-ADPs into interpretable components corresponding to different length scales. New methods validate the global distributions and local values of model B-factors[11,12], but refinement and validation approaches have understandably focussed on optimising the quality of the model, rather than producing an interpretable disorder model that is easily utilisable for structural analysis.

One convenient property of displacement parameters, denoted **U**, is that they are additive, and thus independent contributions can each be represented in the overall disorder model[3]:

$$\mathbf{U}^{\text{total}} = \mathbf{U}^{\text{crystal}} + \ldots$$
$$+ \mathbf{U}^{\text{domain}} + \ldots \qquad (1)$$
$$+ \mathbf{U}^{\text{atom}}.$$

In this work, we present an extensible model that explicitly re-factors the disorder in refined atomic models into a hierarchical series of contributions. This physically motivated formalism separates the different disorder contributions at different scales, as in (1), is generally applicable to both a- and i-ADPs, and enables a quantitative analysis of static/dynamic disorder in macromolecular structures. The disorder identified at different length scales reveals domain motions and loop flexibility, which are likely linked to function, and this approach opens the door to new ways of visualising and understanding macromolecular structures.

## Results

**A hierarchical disorder model**. The **E**xtensible-**C**omponent **H**ierarchical **T**LS (ECHT) B-factor model comprises a hierarchical series of TLS groups that describe disorder at different length scales:

$$\mathbf{U}^{\text{total}} = \sum_{l=1}^{n_{\text{tls}}} \mathbf{U}_l^{\text{tls}} + \mathbf{U}^{\text{atomic}}, \qquad (2)$$

where $n_{\text{tls}}$ is the number of TLS levels in the hierarchical model, $\mathbf{U}_l^{\text{tls}}$ are the TLS disorder contributions for level $l$, and $\mathbf{U}^{\text{atomic}}$ is a set of i-/a-ADPs. The groups on each level are chosen to mirror the hierarchy of physical sources of macromolecular disorder (Fig. 2): lower levels contain large-scale groups, e.g., for each chain or domain, while higher levels contain smaller-scale groups, e.g., for each secondary-structure element, residue or sidechain. For comparison, conventional TLS-refined models (one TLS level with an isotropic atomic component) can be retrieved by setting $n_{\text{tls}} = 1$ and using an isotropic $\mathbf{U}^{\text{atomic}}$.

To parameterise an ECHT model, we utilise an elastic-net-based[13] approach, which assigns disorder to a smaller-scale (e.g., residue) only when it is incompatible with disorder at a larger scale (e.g., molecule). This effectively minimises the complexity of the model required to describe the disorder of a protein and thus produces a parsimonious model of disorder for the structure (see

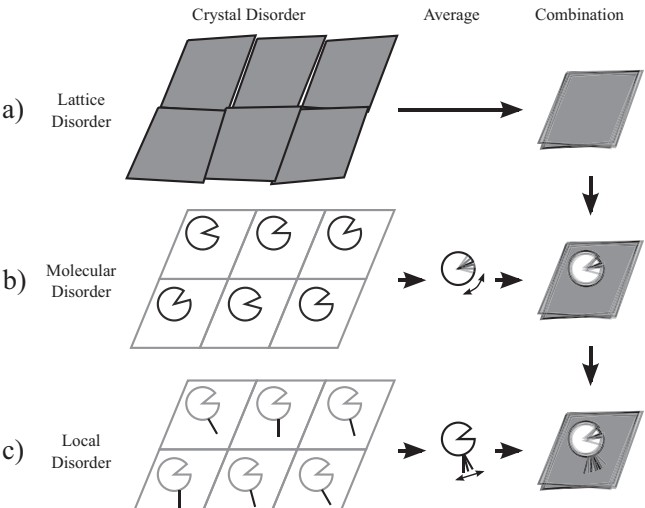

**Fig. 1 Schematic depiction of hierarchical disorder within the crystal.** The three columns show: (left) possible states of the crystal for different scales of disorder; (middle) the average of these states; and (right) the combination of disorder at this level with the disorder from previous rows. Different possible contributions of disorder lead to systematic contributions across **a** the whole unit cell, **b** whole molecules or domains and **c** local elements. Where the disorder terms are uncorrelated, the observed disorder (**c**, right) is simply the sum of these different sources.

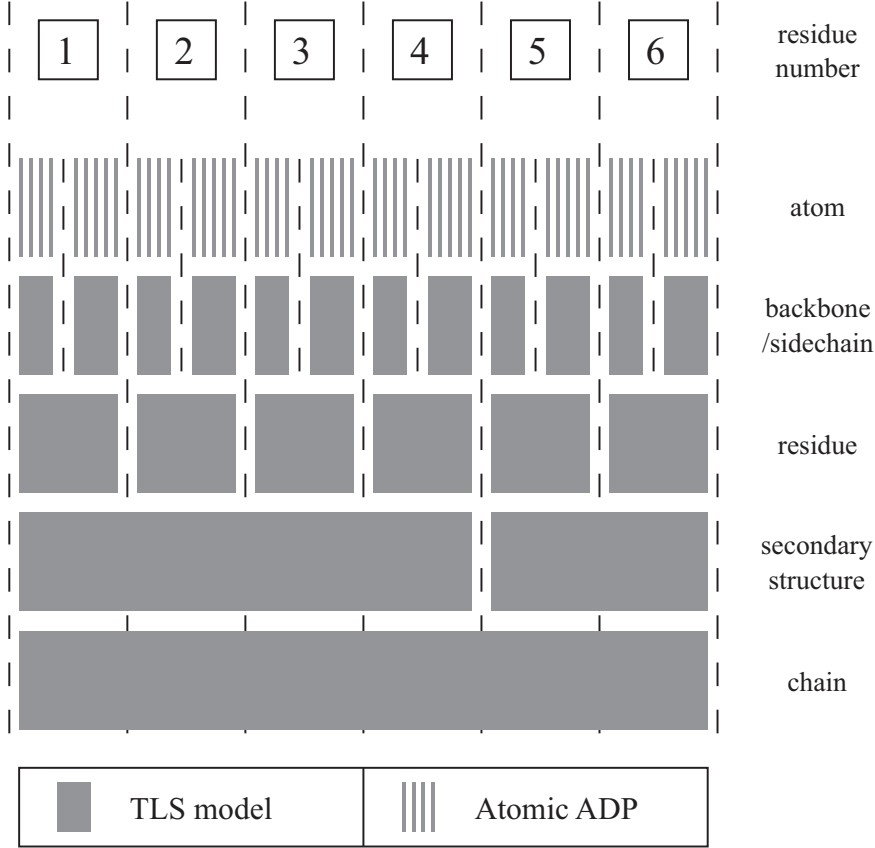

**Fig. 2 Hierarchical disorder model partitioning.** A possible model partitioning is shown for six residues. The bottom (first) level contains a TLS group for all atoms, the second level into two TLS groups, and so on. The final (highest) level contains an ADP for each atom. The total ADP for an atom is given by summing the contributions from each level.

"Methods"). By separating out potentially mutually confounding disorder components, these ECHT decompositions lead to quantitative structural models for molecular flexibility on different length scales.

**SARS-CoV-2 main protease**. To demonstrate the method, we analysed the disorder patterns in a structure of considerable societal importance: that of crystal structure of the SARS-CoV-2 main protease (Mpro; PDBID[14,15] 7k3t; resolution 1.2 Å; refined with individual a-ADPs). The elastic-net optimisation of the ECHT model and the B-factor decomposition profile can be seen in Fig. 3, and show that significant disorder is present at all levels (Supplementary Table 1). Once these mutually confounding components are separated, we identify flexibility in the regions surrounding the catalytic site where structural changes have been observed previously upon substrate binding[16] (Fig. 4). Moreover, the same key components are identified as flexible that are identified in molecular dynamics simulations[16]: the P2 helix, the P5 loop, and the C-terminus. The disorder is also identified at the appropriate level, with the disorder for the P2 helix identified in the secondary-structure level, while for the P5 loop the disorder is principally identified in the residue level.

Another Mpro structure, collected at room temperature, contains a dimer in the asymmetric unit, allowing us to compare the monomers' disorder profiles (PDBID 6xhu[17]; resolution 1.8 Å; refined with TLS and individual i-ADPs; Supplementary Figs. 3–6). Overall, the ECHT components are highly correlated between the two monomers (Supplementary Fig. 7), however some regions display significant differences: the region below the main catalytic site (residues 62–80) is significantly more flexible

in the first monomer (chain A; c.f. Supplementary Figs. 3 and 4), despite the fact that this region forms crystal contacts with symmetry-related molecules in both molecules in the asymmetric unit. When a dimer level is included in the ECHT description, we can extract a collective dimer motion and individual monomer-level rocking motions around the dimer interface (Supplementary Fig. 8).

A third Mpro crystal structure (PDBID 6wqf[16]; 2.3 Å; individual i-ADPs), also collected at room temperature, again displays similar disorder patterns, including the disorder of the region around residues 62–80 (Supplementary Figs. 9 and 10), suggesting that this disorder reflects an intrinsic domain flexibility, which is only visible in certain crystal forms, or at higher temperatures. As this structure was refined with isotropic B-factors, only the magnitude of the anisotropic TLS components are fitted against the input structure's i-ADPs. While residues surrounding the binding site show flexibility comparable to that identified in both 7k3t and 6xhu, 6wqf shows significantly more secondary-structure disorder (Supplementary Table 1).

**SARS-CoV-2 surface glycoprotein**. We subsequently applied the method to another crucial SARS-CoV-2 protein, the surface glycoprotein spike protein, in both closed and open conformations[18] (PDBIDs 6vxx and 6vyb; 2.8 Å and 3.2 Å; individual i-ADPs), which were determined using cryo-EM (Fig. 5). In each of these, the ECHT description contains a level for the whole homotrimer, as well as for each sub-domain (for domain definitions see Supplementary Table 2 and Supplementary Fig. 11), which were identified by manual inspection of the structures. As expected, given the resolution of the structures,

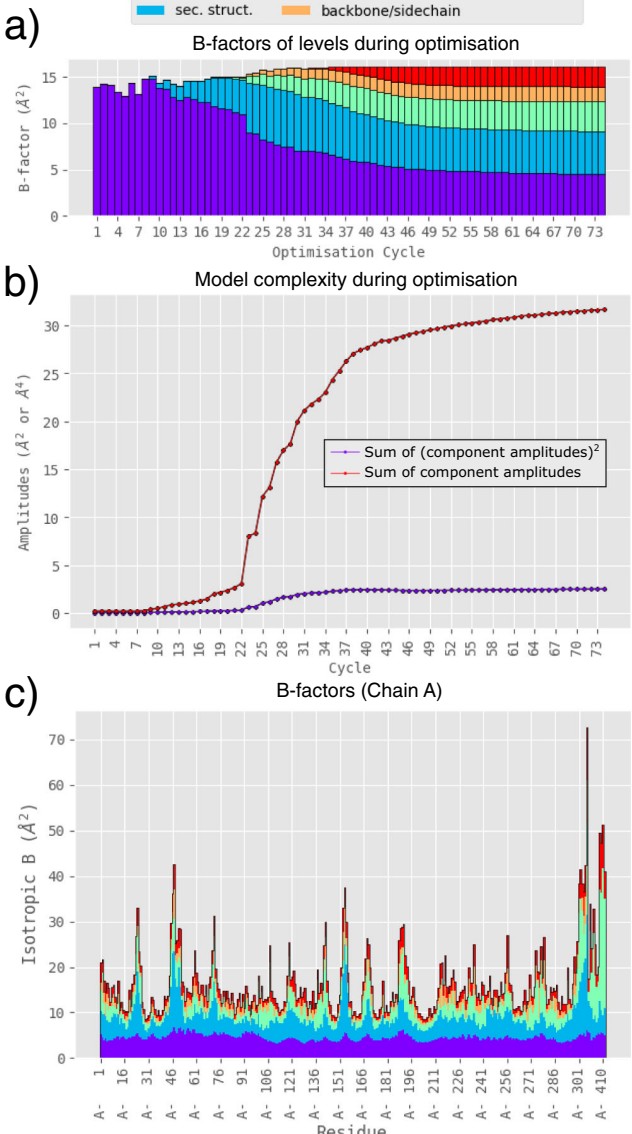

**Fig. 3 Optimisation and ECHT disorder profile of a structure of the SARS-CoV-2 main protease (7k3t). a** The average B-factor of each level for each optimisation cycle. Optimisation begins with large weights on the number of model parameters, forcing disorder to be modelled only using large-scale groups, i.e., at the chain level. As optimisation cycles increase, the penalty on the number of model parameters decreases and disorder is increasingly allowed at smaller and smaller scales, improving model fit until the model converges. **b** The sum of the (squared) amplitudes for each independent model component (e.g., TLS group), representing model complexity, for each optimisation cycle. These values are used to penalise model complexity during elastic-net optimisation. Amplitudes can be converted to the units of B-factors by multiplying by $8\pi^2$. **c** The final ECHT disorder profile after optimisation, shown as the magnitude of the atomic B-factors averaged over each residue and coloured by level.

disorder is assigned mostly to large-scale disorder in the ECHT profile, with the molecule and domain levels accounting for 79% (6vxx) and 82% (6vyb) of the total disorder (Table 1).

The molecular ECHT levels show a gradient of disorder over the model (likely reflecting image alignment uncertainties) and at the tips of the structures, this component masks smaller-scale disorder components. Accounting for and removing this global disorder component allows for interrogation of the region

surrounding the receptor-binding domains (RBDs), which are required for the binding to ACE2 receptors[18]. In the closed conformation (6vxx), the ECHT domain level shows clear hinge-like disorder components that reveal the propensity for RBDs to adopt an alternate conformation. In the open conformation (6vyb), one RBD is observed in an extended conformation, and is highly disordered at both the domain and the secondary-structure level, indicating that the domain is highly flexible. However, the disorder patterns of the remaining two closed RBDs also show significant differences: one RBD (chain C) maintains the distinct hinge-like disorder pattern, while the other (chain A) now displays a smaller, more homogenous, disorder profile at the domain level. The flatter disorder profile of this RBD (chain A) may be a result of a mutually stabilising interaction between this and the extended RBD (chain B). Away from the RBDs, however, equivalent atoms in each part of the trimer demonstrate very similar disorder components at each level (Supplementary Fig. 14).

**STEAP4**. We next analysed the ECHT profiles of two cryo-EM structures of the Six-Transmembrane Epithelial Antigen of the Prostrate 4[19] (STEAP4; PDBIDs 6hcy & 6hd1; resolutions 3.1 Å & 3.8 Å; both refined with one i-ADP per residue). 6hcy was determined from data where $Fe^{3+}$-NTA molecules were present in the protein solution, and although there is evidence of some iron present in the substrate binding site, no iron molecule was modelled[19]; no iron was added in 6hd1. The detail in these disorder models is naturally limited by the resolution and B-factor model, but the ECHT disorder analysis reveals a large number of interesting features. There is a large global disorder component (Fig. 6b), and increased disorder of the globular intracellular domain (Fig. 6c). The homogeneity of the global disorder component differs notably from those of the spike molecules; this difference is potentially explained by the compactness of the STEAP molecules and their embedding within well-defined digitonin mycelles, which leads to a translational component being dominant in the image alignment uncertainty. Additionally, at the secondary-structure level, increased flexibility is clearly visible for the two outermost transmembrane helices (Fig. 6d) and for loops around the substrate binding site (Fig. 6d, e and Supplementary Fig. 17), which is likely necessary for substrate recognition and binding, since $Fe^{3+}/Cu^{2+}$ ions must navigate through rings of alternating positive and negative charges around the binding site to be reduced by the integral HEME molecule[19]. Furthermore, disorder in the secondary-structure and residue levels supports a potential mechanism for electron transport, whereby shuttling of the bound FAD molecule facilitates transfer of electrons (via transfer of a hydride ion) between the NADPH and HEME molecules[19] (Fig. 7). Lastly, the intracellular domains form a tightly ordered trimeric interface, while the other half of the domain is more flexible, suggesting a molten sub-domain whose flexibility may be related to turnover of NADPH molecules bound to the intracellular domain (Supplementary Figs. 19 and 20). All these features are also present in the related structure 6hd1 (c.f. Supplementary Figs. 15–21).

**Comparison of different temperatures**. To further investigate the ECHT decompositions, we analysed 30 pairs of structures collected at room- and at cryogenic temperatures[20], and looked at the changes in disorder patterns associated with the change in temperature. The ECHT decompositions generally reveal an increase in disorder at all scales between the pairs of structures, in line with our expectation that the warming up of the protein affects all scales of motion (Supplementary Fig. 22).

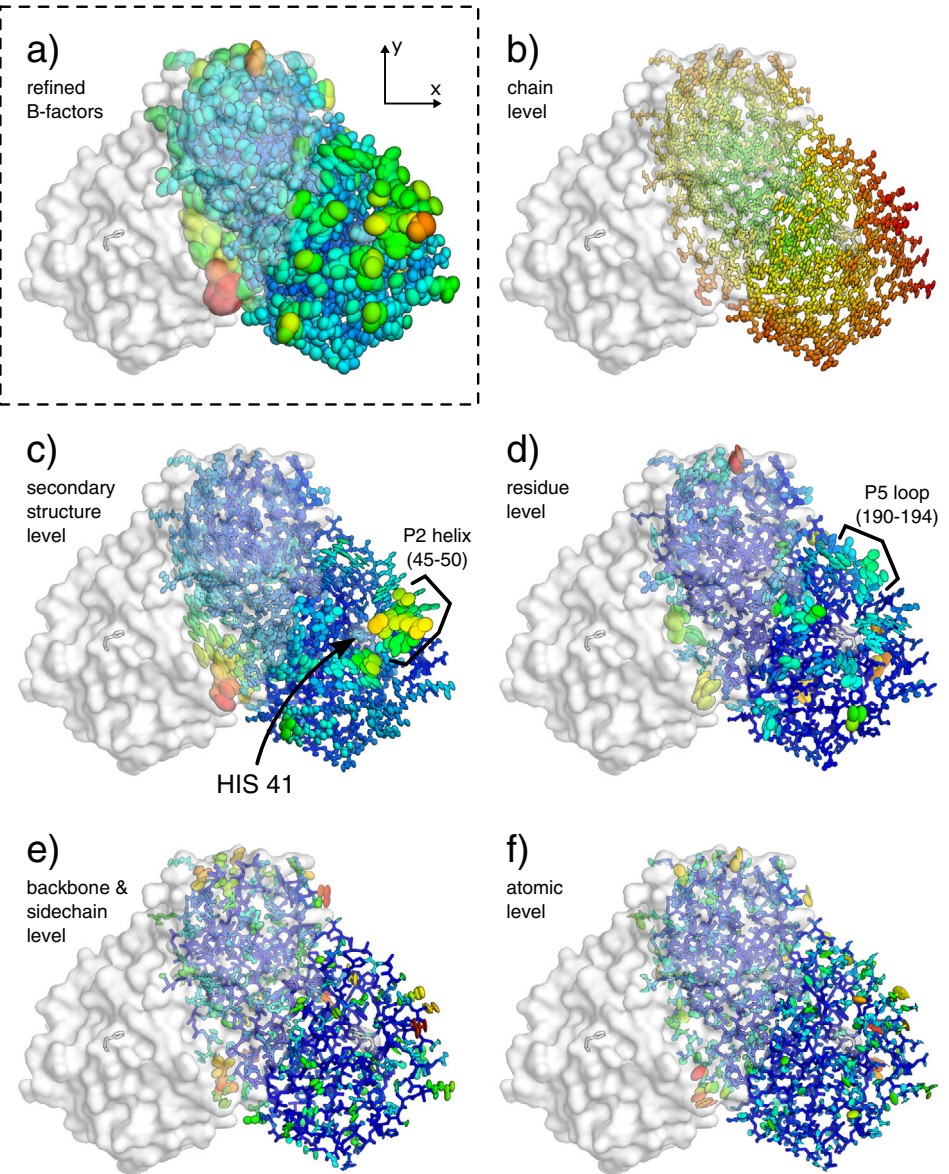

**Fig. 4 ECHT decomposition of a structure of the SARS-CoV-2 main protease (7k3t).** The protein is shown as lines and ellipsoids, coloured by B-factor for each structure independently from blue (zero) to green to red (maximum); the symmetry-related copy creating the obligate homodimer is shown as semi-transparent surface. The binding site histidine (HIS 41) is also shown as a semi-transparent surface, and as sticks in both monomers. B-factor ellipsoids are contoured at $p = 0.95$. **a** Re-refined structure from PDB_REDO[21] (1.2 Å resolution; $R_{work}/R_{free}$ 0.14/0.16; refined with a-ADPs; B-factors 7.7–77.6 Å²). **b–f** Disorder components of each ECHT level (maximum B-factor in brackets): **b** chain (7.4 Å²), **c** secondary-structure (30.2 Å²), **d** residue (35.9 Å²), **e** backbone & sidechain (14.4 Å²) & (**f**) atomic (19.0 Å²). **c, d** Flexible segments and residues line the sides of the catalytic site (see also Supplementary Fig. 1). **c** The P2 helix, on the edge of the catalytic site, shows a large disorder component at the secondary-structure level. The C-terminal also shows a large disorder component (see Supplementary Fig. 2). **d** The P5 loop is also observed to be particularly flexible in the residue level. **e** Backbone motions are generally small, with the majority of disorder being isolated to the surface side chains. **f** Atomic disorder highlights internal motions of residues that cannot be modelled by the rigid-body approximation; these principally highlight longer side chains and backbone carbonyls, as well as presumably absorbing inflated B-factors from modelling errors. Images rendered in pymol[29].

Several notable exceptions show an increase in all levels except the chain level, which decreases. This is most likely due to global differences in the crystals leading to changes in the lattice B-factor component. However, it is also possible that because the parsimonious model reduces the common disorder component to the lowest possible level, and as the disorder profile becomes less homogeneous at higher temperatures, the parsimonious model now requires reassignment of disorder to higher levels to account for distinct motions. This reassignment may necessitate a reduction in the disorder of the chain level in some cases.

## Discussion

By averaging over thousands to billions of molecules, macro-molecular diffraction and microscopy data contain detailed information about molecular dynamics, but this information is obscured by experimental artefacts, and the layers-upon-layers of motions that make the total disorder pattern impossible to interpret in terms of individual components. In this work, we have presented a physically motivated approach for analysing the disorder in an atomic structure by decomposing it into interpretable components. This generalises existing the TLS model to multiple length scales, presents a general framework for analytical

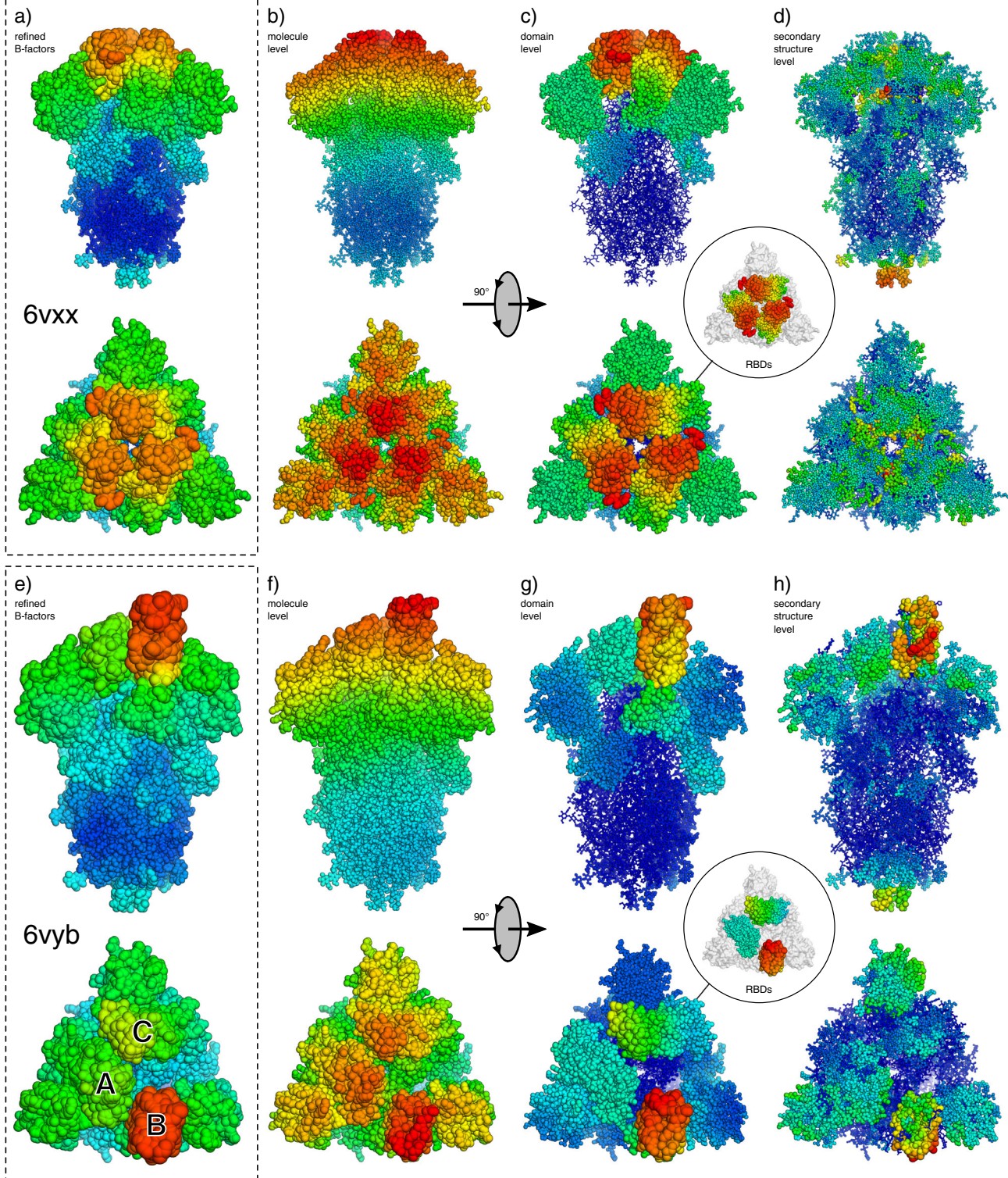

**Fig. 5 ECHT decompositions of SARS-CoV-2 surface glycoprotein structures (6vxx & 6vyb).** Atoms are shown as lines and ellipsoids, coloured by B-factor for each structure independently from blue (zero) to green to red (maximum). B-factor ellipsoids are contoured at $p = 0.95$. Side and top views of trimer for (**a**–**d**) the closed state (6vxx), and (**e**–**h**) the open state (6vyb). **a** Deposited structure for 6vxx (2.8 Å resolution; individual i-ADPs, B-factors 3.5–102.4 Å$^2$). **b**–**d** Disorder components of each ECHT level (maximum B-factor in brackets): **b** molecule (32.0 Å$^2$), (**c**) domain (45.5 Å$^2$), (**d**) secondary structure (20.7 Å$^2$). (**e**) Deposited structure for 6vyb (3.2 Å resolution; individual i-ADPs, 16.4–181.1 Å$^2$). **f**–**h** Disorder components of each ECHT level (maximum B-factor in brackets): (**f**) molecule (67.2 Å$^2$), (**g**) domain (85.3 Å$^2$), (**h**) secondary structure (44.3 Å$^2$). **b**, **f** The molecular components for both structures show similar profiles, with a gradient of disorder across the molecule. **c** The domain-level disorder reveals hinge-like components for the closed RBDs. **g** The domain-level disorder reveals conservation of the hinge-like disorder for chain C, but a loss of this component for chain A. Inset images show the disorder for the RBDs only. **d**, **h** The secondary-structure level reveals intra-domain flexibility. Images rendered in pymol[29].

**Table 1 Distribution of B-factors for ECHT decompositions of SARS-CoV-2 surface glycoprotein (6vxx & 6vyb). The columns contain the average B-factors for each ECHT level in both Å$^2$ and as a percentage of the total B-factor.**

| Level | 6vxx (closed) Average B (Å$^2$) | 6vyb (open) Average B (Å$^2$) |
|---|---|---|
| Molecule | 12.4 (43.1%) | 26.4 (55.1%) |
| Domain | 10.3 (36.0%) | 12.9 (27.0%) |
| Secondary structure | 3.9 (13.4%) | 5.8 (12.1%) |
| Residue | 1.7 (5.8%) | 2.1 (4.3%) |
| Atomic | 0.5 (1.7%) | 0.7 (1.5%) |

disorder decomposition that applies for both isotropic and anisotropic disorder, and enables quantitative structural analysis. Whilst not a viable method for structure refinement as presented here, hierarchical disorder models have several advantages that may lead to future improvements in the refinement of B-factors.

The advent of cryo-EM raises the exciting opportunity to study molecules outside the context of a crystalline environment, but there clearly remains an opportunity to study macromolecular dynamics in crystals. Both large-scale and small-scale disorder contain useful information for structural analysis: large-scale molecular disorder dictates how well molecules are resolved and reveal domain motions (Fig. 5); secondary-structure-level disorder helps form hypotheses for structural mechanisms (Fig. 6); and small-scale disorder details the rigidity/plasticity of binding sites (Fig. 4). Since this disorder is now characterised on an absolute scale (describing the fluctuations of an atom in Å), the relevant components can be used in a variety of quantitative structural and bioinformatic applications.

Naturally, the output ECHT model and a subsequent analysis are dependent on the choice of model levels, and how these levels are partitioned (e.g., choice of secondary structure). Groups must therefore be chosen carefully and critical analysis of the model is essential. However, we have found that starting with the default levels—chain, (automatically identified) secondary structure, residue, backbone/sidechain, and atomic levels—generally produces reliably interpretable results. If an analysis with these levels reveals large stretches of residues with similar disorder patterns (e.g., domains, formed of neighbouring secondary-structure components), new level groupings can be identified, as in the case of the spike protein structures (Fig. 5). These new groupings then serve as a hypothesis for a further analysis: if collective disorder patterns for the group are not present, then these new groupings will contain a negligible disorder component in the output model, implying that collective motion is unlikely. To assist users in using the method, case studies and worked examples will be made available (see availability).

Conversely, at lower resolutions, one can safely remove finely partitioned levels (e.g., the backbone/sidechain level), since the model will not be expected to contain information at these length scales, as per the STEAP analyses. However, including overly detailed levels will not generally cause problems, since the elastic-net method will suppress unnecessary groups that are not needed to reproduce the refined B-factors. The method is therefore relatively robust to over-parameterisation: examples in this work demonstrate this well (Table 1 and Supplementary Tables 1 and 3), where the atomic level only has a non-negligible component for the higher-resolution structures.

The parametric redundancy of the ECHT model means that in the general case there are the same number of independent parameters in the input refined structure and the output fitted ECHT model. Because of this one-to-one relationship between the input and output models, the quality of the input model and the refinement protocol have a strong effect on interpretation: regions of particular interest in a structure should be checked to ensure the refined model is of high quality with negligible difference density. Additionally, overly strong B-factor restraints must be avoided, which might cause atoms to have overly similar disorder profiles, thereby channelling disorder to larger scales than the data might otherwise suggest. Fortunately, overly strong B-factor restraints will reflect negatively in R-free values, so well-refined structures should generally produce good results. The testing of multiple B-factor restraint weights by automated re-refinement pipelines such as PDB_REDO[21] makes them highly recommended for generating unbiased input models for ECHT analyses. Validation tools can also be used to validate the quality of the input model B-factors[11,12], and automated integration of these tools is a clear avenue for future work. Additionally, the quality of the ECHT model fit can be visualised by looking at the atomic level, as noise in the refined atomic B-factors will appear as random atomic components in the atomic level; explicit quantification of a noise level from non-physical fluctuations in the atomic-level provides another avenue for future work.

Because of the parametric redundancy between the different levels, the complexity of the ECHT model is not measured by the number of non-zero parameters, as is the case for most B-factor models. Instead, a measure of complexity for an ECHT model can be calculated as the sum of the model component amplitudes, as shown in Fig. 3 (see "Methods" for more details). This value can be directly compared between different ECHT analyses of the same model, but a normalised complexity can also be calculated by dividing this number by the number of atoms in the model. This value may eventually provide a way of quantifying the information in a set of model B-factors.

Naturally, a disorder component at a particular level does not necessarily imply correlated motion, only that atoms have compatible disorder profiles; correlated motion can only be confirmed by orthogonal methods, such as diffuse scattering[22,23]. Furthermore, when optimising an ECHT model against models refined with i-ADPs, multiple TLS parameter sets may provide a similar fit to the input i-ADPs. Currently, visual analysis of the output model is necessary to ensure it is physically reasonable, and does not contain large changes in the magnitude or direction of disorder between adjacent residues. The addition of appropriate model restraints to enforce these restrictions, as well as the implementation of symmetry constraints, remains an avenue of future work. The use of TLS-described rigid-body motions is also a simplification of the continuous motions that proteins will undergo in reality, however, we have shown here that layers of rigid-like components can still be used to gain insight into the magnitude and localisation of macromolecular disorder, and that TLS models are more than merely a useful mathematical formalism. The elastic-net optimisation approach is also naturally applicable to any other disorder formalism that may be developed.

The approach presented here is a general and easily applicable analysis tool that provides a refactoring of the structural disorder into layered riding motions, and allows disorder analysis to become a standard step during macromolecular structural studies. This intuitive remodelling of disorder allows physically plausible interpretations of macromolecular disorder and enables a range of studies related to the understanding, alteration, and manipulation of macromolecules.

## Methods

**Input structures**. The ECHT disorder model is fitted to a refined atomic structure. Input structures can be parameterised with either i-ADPs or a-ADPs, or a mixture.

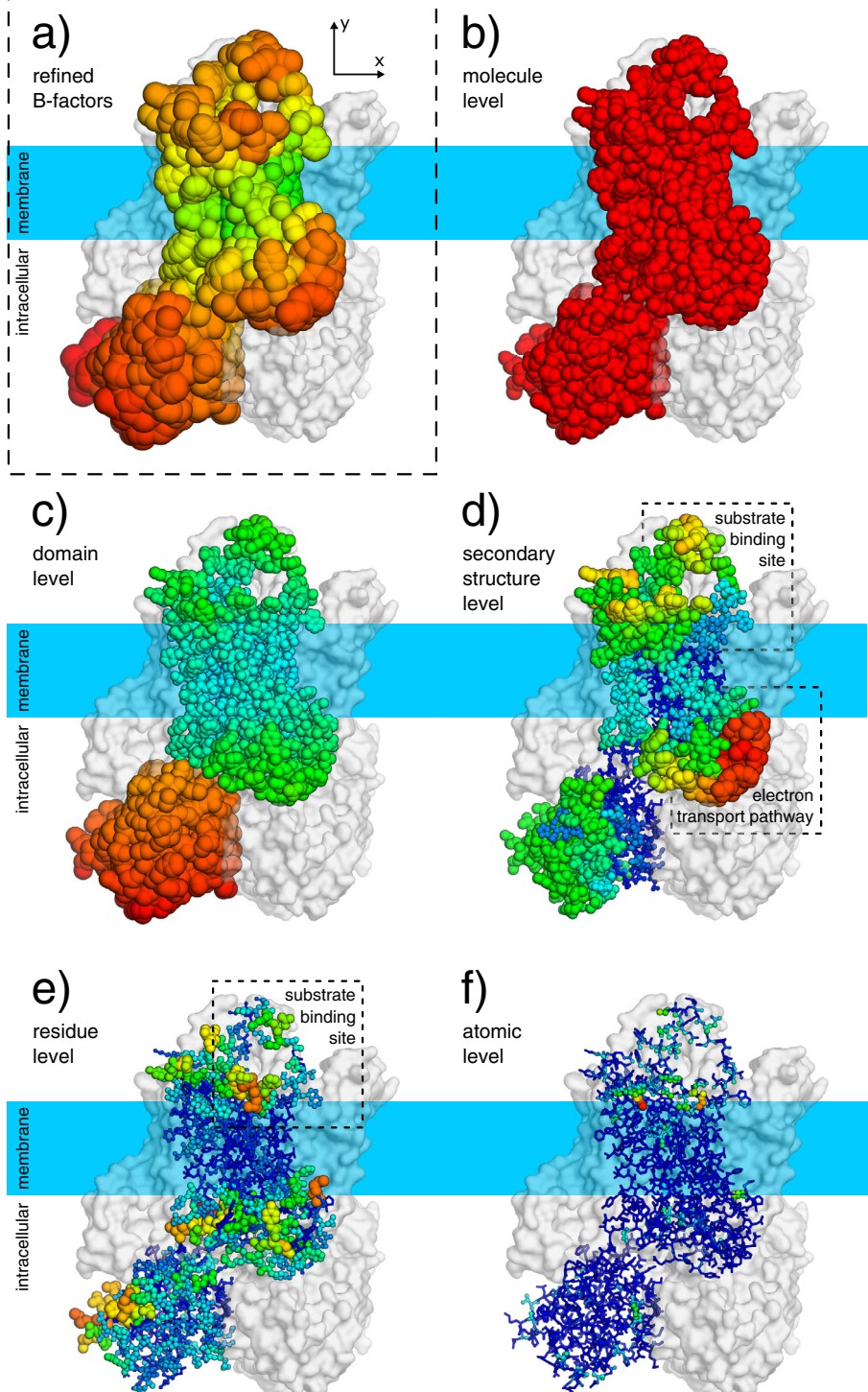

**Fig. 6 ECHT decomposition of a STEAP4 structure (6hcy).** Atoms are show as lines and ellipsoids, coloured by B-factor for each structure independently from blue (zero) to green to red (maximum). B-factor ellipsoids are contoured at $p = 0.95$. **a** Deposited structure (3.1 Å resolution; one i-ADP per residue; B-factors 55.4–146.7 Å$^2$). **b–f** Disorder components of each ECHT level (maximum B-factor in brackets): **b** molecule (44.7 Å$^2$), **c** domain (65.8 Å$^2$), **d** secondary structure (51.7 Å$^2$), **e** residue (28.7 Å$^2$), & (**f**) atomic (13.8 Å$^2$). The backbone/sidechain level was excluded due to resolution and the group i-ADPs; the atomic level was included to stabilise the optimisation, and has a negligible disorder component (see also Supplementary Table 3). For domain definitions see Supplementary Table 4. **b** The global disorder component is largely isotropic. **c** The extracellular domains display increased flexibility relative to the membrane-embedded domains (d) The two outermost transmembrane helices are significantly more disordered than the two inner helices. **d, e** The loops surrounding the substrate/co-factor binding sites display significant flexibility, which suggest links to function. Images rendered in pymol[29].

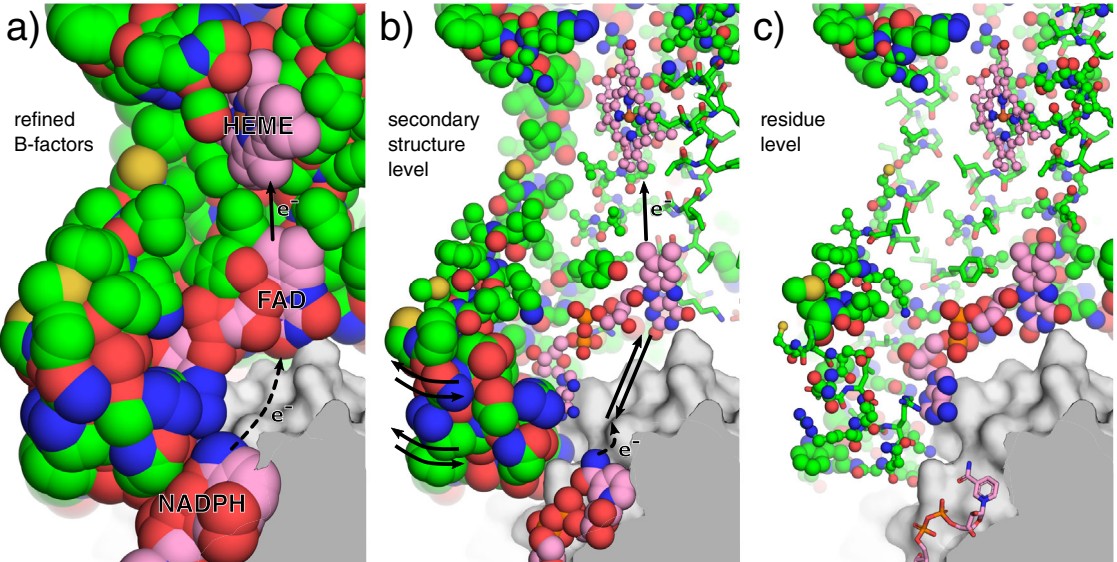

**Fig. 7 Putative flexibility-based mechanism of electron transport in STEAP4.** Cutaway view of a wing of the STEAP4 molecule (6hcy). Representation as in Fig. 6 except for colour: for chain A, protein carbon atoms are shown in green, and non-protein carbon atoms are pink; atoms are otherwise coloured by element. **a** Deposited B-factors. The electron transport pathway in STEAP4 requires the transfer of a hydride ion from the bound NADPH molecule to the FAD molecule, which subsequently transfers two electrons to the heme molecule, which subsequently reduces bound substrate molecules, $Fe^{3+}/Cu^{2+}$ ions. The pathway for transfer of the hydride from the NADPH molecule to the FAD molecule is unknown[19]. **b** ECHT secondary-structure level. The observed flexibility suggests a mechanism whereby shuttling of the FAD molecule, caused by breathing of the flexible wing, brings the FAD flavin moiety in close proximity to the NADPH, where it can gain the hydride, and then subsequently transfer electrons to the heme molecule upon returning to the solved state. **c** The residue level reveals that the FAD molecule has increased disorder relative to its surroundings, further suggesting that structural transitions could be possible. Images rendered in pymol[29].

For TLS-refined structures, the TLS contribution can either be included in the atomic ADPs (resulting in a-ADPs), or excluded (resulting in i-ADPs).

**Hierarchical model partitioning.** Default TLS group partitions are created for each: protein chain; local secondary-structure element; residue; and residue backbones/side chains. Secondary structure is automatically identified with the DSSP algorithm[24] as implemented within the cctbx[25]; $C_\beta$ atoms are included with backbone atoms. Glycine, Alanine and Proline residues are not considered in the backbone/sidechain levels. An atomic level is created with an a-ADP or i-ADP (depending on the input disorder model) for each atom. Part of an example composition is shown in Supplementary Fig. 24. All ECHT levels mentioned in this work use the above model composition unless stated otherwise. Custom levels can also be defined, such as for individual domains within a chain, or across chains. To aid interpretability, levels should generally be ordered such that the largest groups appear at lower levels.

**ECHT model parameterisation.** The overall schema of the parameterisation process is shown in Supplementary Fig. 23. In each macrocycle, the algorithm alternates between optimising the TLS parameters for each level using a simplex-based approach, and optimising the amplitudes of all model components (TLS groups for TLS levels, and ADP values for the atomic level), using a gradient-based method with elastic-net penalties. Between the macrocycles, elastic-net penalties are decayed by a chosen factor (between 0 and 1). Smaller values lead to a faster runtime but may miss model components (e.g., lead to a non-parsimonious model); large values lead to a longer runtime, but a much more detailed model. In this work, the decay factor is chosen to be 0.8, which was found to be a good balance between speed and model quality. Some minor differences may appear between symmetry-related monomers in a model; these can be a good indicator that the decay factor is too small. Within each macrocycle, each series of TLS and amplitude optimisations is repeated until the amplitudes of the components stop changing between each microcycle.

*Target function.* All optimisation uses a least-squares target function,

$$\sum_a \omega_a \cdot (\mathbf{U}^{\text{target}} - \mathbf{U}^{\text{optimise}})_a^2, \tag{3}$$

where $\omega_a$ is the weight for atom $a$, $\mathbf{U}^{\text{target}}$ are the optimisation target ADPs, and $\mathbf{U}^{\text{optimise}}$ are the ADPs being optimised. For amplitude optimisation, the input ADPs, $\mathbf{U}^{\text{input}}$, are used as $\mathbf{U}^{\text{target}}$. For the optimisation of individual TLS group

parameters, the $\mathbf{U}^{\text{target}}$ values for level $k$ are given by

$$\mathbf{U}_k^{\text{target}} = \mathbf{U}^{\text{input}} - \sum_{l \neq k} \mathbf{U}_l^{\text{echt}}, \tag{4}$$

where $\mathbf{U}_l^{\text{echt}}$ are the ADP values for level $l$ of the ECHT model, and the sum is over levels not currently being optimised.

In this work, the weights used are

$$\omega_a \propto \frac{1}{\frac{1}{3}\text{Tr}(\mathbf{U}_a^{\text{input}})}, \tag{5}$$

which reduces the effect of atoms with large B-factors on the optimisation of large-scale levels—for which they contain little information—during the initial cycles. Input weights are normalised such that

$$\sum_a (\omega_a) = 1. \tag{6}$$

*TLS group normalisation.* The $\mathbf{U}$ values arising from a TLS group are rewritten as

$$\mathbf{U}_g^{\text{tls}} = A_g^{\text{tls}} \cdot \widehat{\mathbf{U}}_g^{\text{tls}} \tag{7}$$

for group $g$. The scaling between $A_g^{\text{tls}}$ and $\widehat{\mathbf{U}}_g$ is chosen such that

$$\frac{1}{n_g^{\text{atoms}}} \sum_{a \in g} \frac{1}{3} \text{Tr}\left( \left[ \widehat{\mathbf{U}}_g^{\text{tls}} \right]_a \right) = 1, \tag{8}$$

where the sum $a \in g$ is over all atoms that belong to group $g$, and $n_g^{\text{atoms}}$ is the number of atoms in group $g$. This normalisation scales the TLS-matrix values so that the amplitudes $A^{\text{tls}}$ are proportional to the average B-factor of each group. Note, however, that an amplitude of $A_g^{\text{tls}} = 1 \text{Å}^2$ corresponds to an average B-factor of $8\pi^2 \text{Å}^2 \approx 72 \text{Å}^2$ for this group.

*TLS-level optimisation.* TLS matrices are initialised with an isotropic T-matrix, $diag(1, 1, 1)$, and zero-value L-, and S-matrices. Amplitudes are initially set to zero, and become non-zero during amplitude optimisation (see below). TLS-matrices are optimised using a simplex search method implemented in cctbx[25]; the starting simplex for optimisation of the matrix values is obtained by independently incrementing each normalised matrix component (6 T-elements, 6 L-elements, 8 S-elements) in the coordinate basis of the L-matrix. During simplex optimisation, invalid combinations of TLS-matrix parameters are rejected using the TLS-decomposition protocol described in Urzhumtsev et al.[9]. After matrix optimisation, the TLS matrices and amplitudes are renormalised according to (8).

*Atomic-level optimisation.* Atomic-level optimisation is performed using the same least-squares target and simplex search method as the TLS levels with the

constraint that the resulting ADP is positive semi-definite to within some small tolerance $\epsilon$.

*Inter-level amplitude optimisation.* After each component optimisation, the magnitudes of all TLS groups and atomic ADPs are optimised using the lbgfs algorithm implemented in cctbx[25], with the constraint that no amplitudes can be negative. The amplitudes for TLS groups are defined in (8). The amplitudes of each individual ADP are defined similarly as $A_a^{\text{atom}} = \frac{1}{3}\text{Tr}(\mathbf{U}_a^{\text{atom}})$; the atomic normalisation then becomes

$$\mathbf{U}_a^{\text{atom}} = A_a^{\text{atom}} \cdot \widehat{\mathbf{U}}_a^{\text{atom}}. \tag{9}$$

To minimise the model complexity needed to describe the disorder, the following elastic-net penalties[13] are added to the target function:

$$\Omega^\alpha \cdot \left[ \sum_g A_g^{\text{tls}} + \sum_a A_a^{\text{atom}} \right] \tag{10}$$

and

$$\Omega^\beta \cdot \left[ \sum_g (A_g^{\text{tls}})^2 + \sum_a (A_a^{\text{atom}})^2 \right], \tag{11}$$

where $\Omega^\alpha$ and $\Omega^\beta$ are the lasso (sum of amplitudes) and ridge-regression (sum of squared amplitudes) weights, respectively. These can be redefined in terms of a mixing parameter

$$\gamma = \Omega^\alpha / (\Omega^\alpha + \Omega^\beta) = \Omega^\alpha / \gamma_0, \tag{12}$$

where $\gamma = 1$ is lasso regression and $\gamma = 0$ is ridge-regression. Larger values of $\gamma$ bias towards the parsimonious model, while smaller values lead to more non-zero components. In this work a mixing value of $\gamma = 0.9$ is used.

At the end of every optimisation macrocycle, the elastic-net weights are reduced by a factor $\delta$:

$$\Omega_{n+1} = \Omega_n * \delta \tag{13}$$

where $n$ is the macrocycle number and $\delta$ is a weight decay factor between 0 and 1.

**Convergence**. Optimisation cycles continue until all ECHT model component amplitudes are changing by less than $\Delta B_{\text{cutoff}}$ over a fraction of the total cycles. Additionally, a cutoff can be given that terminates the optimisation when a threshold is reached for the RMSD between the $\mathbf{U}^{\text{input}}$ and $\mathbf{U}^{\text{echt}}$. Note that when the atomic level is not included in the optimisation, there is no guarantee that a given RMSD cutoff will ever be reached.

**Program output**. The implementation outputs the disorder contributions for each level as separate structures. A large number of analytical graphs are automatically generated to allow the visual analysis of the disorder; these are combined into one html results file for ease of use.

## Data availability
All data presented are available on Zenodo[26] (https://doi.org/10.5281/zenodo.5082172); this contains all data required to regenerate the figures and graphs included in this work.

## Code availability
This method as described above is implemented within the PanDEMIC project (Pan-Dataset Ensemble Modelling of Iso-structural Crystals; named prior to the ongoing coronavirus pandemic) and the script *pandemic.adp* and is available within the *panddas* python package (https://pandda.bitbucket.io, https://bitbucket.org/pandda/pandda). The program will be distributed within CCP4[27], but can equally be installed within a phenix[28] environment with an up-to-date cctbx installation[25].

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

## Acknowledgements
N.M.P. recognises funding from a long-term fellowship from the European Molecular Biology Organisation (EMBO), fellowship number ALTF 609-2017, and from a Veni Fellowship, fellowship number VI.Veni.192.143 from the Dutch Research Council (NWO). P.G. recognises funding from the Dutch Research Council (NWO), project number 01.80.104.00. N.M.P. thanks Helen Ginn, Billy Poon, and James Parkhurst for their help and guidance with c++ and cctbx. N.M.P. and P.G. thank Wout Oosterheert for discussions of the STEAP structures, the coronavirus structural task force (inside-corona.net) for discussions of the SARS-CoV-2 structures, and Loes Kroon-Batenburg,

Tim de Klijn and Jitse van der Horn for general discussions regarding disorder. N.M.P. is especially grateful to Arwen Pearson for her invaluable and indefatigable support.

## Author contributions

N.M.P. and P.G. designed the research and wrote the manuscript. N.M.P. developed and implemented the algorithm, and performed all analyses.

## Competing interests

The authors declare no competing interests.
