## [Peer Review File · Nature Communications]

A method for intuitively extracting macromolecular dynamics
from structural disorderREVIEWER COMMENTS

Reviewer #1 (Remarks to the Author):

The manuscript submitted by Pearce and Gros describes a new protocol for protein crystal structure refinement. The principal aim of this protocol is the computation of B-factors that monitors genuine atomic motions around atomic average positions – oscillations, transitions from a conformation to another, etc. – not influenced by other factors – crystal defects, large rigid-body motions of structural moieties, etc. Software is distributed – within CCP4 and other software packages – to allow users to adopt this refinement strategy.

The core of this refinement protocol, named Extensible-Component Hierarchical TLS (ECHT) B-factor refinement, is a series of TLS group partitions, which are somehow hierarchical: (i) protein chain, (ii) individual domains within a chain (optional, only when needed) (iii) secondary structure element, (iv) residue, and (v) backbones-side chains.

The examples, given to illustrate the potential of ECHT, are insufficient to prove the general applicability of this refinement protocol. It is necessary that as many crystallographers as possible test ECHT to really verify its performance. This is, in my opinion, a sufficient reason to publish this manuscript, not necessarily in this journal – I would recommend a crystallographic journal.

Detailed comments:

(i) The first sentence of the introduction is a cliché that might be better reproduced as “Macromolecular crystallography records a temporal average – diffraction data collection requires time – and spatial average – multiple copies of the molecule are present in a single crystal”.

(ii) The description of crystal lattice defects and local conformational disorder is pretty naive. There are numerous books devoted to molecular crystals (the classic *Molecular Crystals* by Wright, for example) with chapters focused on these topics. Even Wikipedia is better. I suggest the Authors better describe this part.

(iii) Page 2, left column, bottom. The sentence “One convenient property of displacement parameters, U ,” mentions U , which has never been previously defined.

(iv) In the Discussion – page 8 right column – the Authors correctly write that “The output ECHT model is naturally dependent on the choice of levels, and how these levels are partitioned (e.g. choice of secondary structure). Therefore, groups must be chosen carefully and critical analysis of the model is essential.” This is absolutely true and crucial. Would it be possible to provide some strategies to “choose carefully” the groups to be analysed?

(v) It might be interesting to compare ECHT results computed on the same crystal structure (same protein, same space group, same pH, etc.) determined at different resolutions. Actually, one can imagine that ECHT behaves and might be used differently at different resolution levels.

Reviewer #2 (Remarks to the Author):

I consider this project and the results presented in the manuscript as extremely significant to the field of structural biology.

This is a kind of projects where a number of people may say “I was talking about this years ago...” but where such declaration means nothing by itself. The key result of this work is that the authors, first time for macromolecular structural studies, developed and practically realised a hierarchic model describing mobility of macromolecular structures. As the authors show by their examples, this both

gives fundamental description of such mobility and suggests its role in molecular mechanisms (STEAP4, Fig. 7). The computer tools developed, being distributed within the major suites such as ccp4 and phenix, will allow the community of structural biologists to get a deeper understanding of structure-function relations.

This key 'ECHT method and program' result should not hide another key issue, a long-time controversy if the TLS models indeed have some physical meaning or if the respective matrices are only formal 'dummy' parameters. Excellent examples presented in this work strongly support the former point.

I wonder if this method also may be used to validate atomic displacement parameters, ADPs. Their refinement seems to be an established procedure in X-ray crystallography but yet not so well in cryo-electron microscopy (cryo-EM). One can consider the models of SARS-Cov-2 surface glycoprotein as an example. In addition to the entries used in this work (6vxx and 6vyb), PDB contains at least 7 other entries (6moj, 6wpt, 6xey, 6zb5, 6zp2, 7a97, 7kj2). In these models, the same residues have B-factors that vary from 20 to 400 Å². I believe that the ECHT approach can answer what is a physical significance of these ADPs, if any.

By all these reasons I expect that the scientific community will be happy to have an access, as soon as possible, to these results, both to the theory and the software.

An important feature of this method is that it does not necessary require high-resolution data and is applicable to structures solved at a resolution that varies in a quite large range. Also, the examples prove that the method is applicable to atomic models obtained by any of the two experimental techniques, crystallography or cryo-electron microscopy, naturally under condition that the refinement of ADPs has been done correctly.

The numerous Pymol-figures illustrate the decomposition of the atomic motion into the components of a different level. At my point of view, the authors could comment more on Figure 3a. It shows that, after a few cycles at a new level, the respective ADP contribution practically does not change anymore, and the source for a next level is always the "chain" one. I think this may be considered as an extra validation of the parameters, their robustness: one may interrupt the procedure at any level and be sure that the parameters at this and previous levels are correct or near correct (except "chain" that yet includes a contribution from non-analysed levels).

The authors know well the literature in the field and efficiently use available algorithms, software and methods, with appropriate references to them. Their results, presented by tables, figures and explained in text, support the conclusions and claims done by the authors. The method seems to be quite straightforward and therefore robust (the principal parameter is the weight for the elastic net penalties; am I right? Are there other parameters? – if yes, the authors need to clarify them).

From my point of view, there are still several points where the work and the manuscript can be improved. As for other methods, there are questions: what to do 'in principle'? how to do this practically? how to validate the results? My main concern is the latter.

In the manuscript, the authors, experts in the field, validate their calculations by a careful structural inspection of the results. Since the tool certainly will be largely used, I may imagine a colleague who has no such large experience as the authors and who wished to use ECHT. How he / she can validate their results? How they may be sure that there is no over-ECHT-interpretation of the available ADP? I would suggest the authors of the manuscript to work more on the question of quantitative and objective validation of the ECHT results. How many data they have (Natoms in case of isotropic ADP or 6*Natoms in case of anisotropic ADPs, plus data from the elastic penalties, right?). How many parameters appear at each level (give the numbers for the examples considered in the manuscript)? It may be useful to remind a reader that the atomic coordinates are no longer adjustable parameters so one should not count them as unknowns. By the way, the authors should correct the phrase between formula (9) and (10); the elastic penalties do not minimise the number of parameters but increase the number of 'data'.

Should the decomposition into levels stop as soon as the number of parameters becomes larger than the number of data, if this happens? Should it stop if the number of added parameters results only in a tiny modification of ADP, say, below 1% or is it indirectly included into the current conditions (section 'Convergence')?

It would be great to give some statistical number that takes into account both the closeness of the

UECHT to Uinput and the number of parameters.

I have a few more small comments, mostly technical.

- As I learned, cryo-EM models typically have a larger disorder at their periphery due to uncertainties in the orientation of projections (“molecule level”). It looks like there is no such effect in 6hcy (Fig. 6). Could the authors comment more on this ?
- In section “Methods”, the authors use U for tensors (matrices in a given basis). What do they mean by $|U|$ in formula (5)? Its determinant? In formula (3), do they mean a (hidden) sum over the matrix elements? – these notations should be improved.
- I expect that the coordinate basis of the L-matrix (section ‘TLS-level optimisation’) may vary with iterations; is it important for convergence ? Any comment?
- It might be easier to follow the text if the principal section “Applications to Structural Analysis” has any kind of separation between the three examples.
- Is it correct that His41 is highlighted twice : by a transparent surface (hardly seen) in a right copy AND, conversely, by sticks in the left copy? Should this be clarified?
- In (very) extended Supplementary Materials, figure captions 2, 4, 5, 10 refer to y-axis, and Fig 17 refers to x-axis while neither of these axes is indicated. Either the axes should be plotted or these phrases should be corrected.
- I am unhappy with the presentation in Supplementary Figs. 7, 13, 14. I understand the idea of the authors, however giving ‘grey’ uninformative distributions and making each ‘black’ distribution in two (symmetric) copies does not give a better understanding but rather confusing and these figures become in total too expanded. I suggest that the authors rethink and optimise these figures. This will allow them to increase the size of each individual informative ‘black square’.

My last query, which is below, may be ignored. However, since the authors act in the frame of ‘pandemic’ and since they already have the most of data in their hands, an answer on this query could be interesting, could complete the results of their project (unless the authors develop it as a small supporting opportune project) and be useful practically.

Site <https://en.wikipedia.org/wiki/EpiVacCorona> tells about one of the Covid-19 vaccines, and a Russian version of the same site, <https://ru.wikipedia.org/wiki/ЭпиВакКорона>, shows the components of this vaccine (see also sites <https://patenton.ru/patent/RU2738081C1> and https://yandex.ru/patents/doc/RU2738081C1_20201207). The first component corresponds to the residues 454-477 of SARS-Cov-2 surface glycoprotein. These residues seem to be highly mobile and even missed in some of the nine models of this protein available in PDB (see above). Do the authors see anything particular in their results concerning this part of the protein? Maybe, some other models can give more information, such as 6m0j (crystal; resolution 2.45 Å), 6zp2 (resolution 3.10 Å) or 6zb5 (resolution 2.85 Å) ?

Alexandre Urzhumtsev

REVIEWER COMMENTS

We have numbered our changes and marked these changes in the manuscript.

Reviewer #1 (Remarks to the Author):

The manuscript submitted by Pearce and Gros describes a new protocol for protein crystal structure refinement. The principal aim of this protocol is the computation of B-factors that monitors genuine atomic motions around atomic average positions – oscillations, transitions from a conformation to another, etc. – not influenced by other factors – crystal defects, large rigid-body motions of structural moieties, etc. Software is distributed – within CCP4 and other software packages – to allow users to adopt this refinement strategy.

Regrettably, it seems that the reviewer has misunderstood the purpose of the manuscript, since we present an analytical B-factor model, not a new protocol for B-factor refinement.

(change 1) We have added clarifying statements in the abstract (added ‘analytical’), introduction, and discussion section to clearly state the scope of the manuscript, including an explicit statement that this model is not a refinement protocol.

The core of this refinement protocol, named Extensible-Component Hierarchical TLS (ECHT) B-factor refinement, is a series of TLS group partitions, which are somehow hierarchical: (i) protein chain, (ii) individual domains within a chain (optional, only when needed) (iii) secondary structure element, (iv) residue, and (v) backbones-side chains.

The examples, given to illustrate the potential of ECHT, are insufficient to prove the general applicability of this refinement protocol. It is necessary that as many crystallographers as possible test ECHT to really verify its performance. This is, in my opinion, a sufficient reason to publish this manuscript, not necessarily in this journal – I would recommend a crystallographic journal.

We chose to limit ourselves to a limited number of examples to allow an in-depth discussion of the complexity of the dynamics. However, to show the range of applicability of the method, we chose our examples to include structures of different sizes, determined at a range of resolutions and refined with different refinement protocols, and determined by both crystallography and cryo-EM.

Indeed, we agree that this analytical method should be applied by as many crystallographers and microscopists as possible. Such a wide-spread use may result in a marked impetus for studying the complex dynamics-function relationship in biomacromolecular structure, which is why we think a general journal is appropriate.

Detailed comments:

(i) The first sentence of the introduction is a cliché that might be better reproduced as “Macromolecular crystallography records a temporal average – diffraction data collection

requires time – and spatial average – multiple copies of the molecule are present in a single crystal”.

We agree with the reviewer that our statement borders on a cliché, but it is the most appropriate, accurate, and concise statement of what we are trying to achieve. The replacement that the reviewer suggests, though correct, loses some of the subtleties of the situation – namely that in crystallographic averaging, a spatial average can be the same as a temporal average (i.e. time and space can be interchangeable). Therefore, breaking the statement with clarifications of what is being averaged removes this ambiguity.

(2) *We have modified the beginning of the introduction to merge the two sentences to remove some of the elements of the cliché, and to explicitly link the first statement to the second, which was our initial intention that did not come across strongly enough.*

(ii) The description of crystal lattice defects and local conformational disorder is pretty naive. There are numerous books devoted to molecular crystals (the classic Molecular Crystals by Wright, for example) with chapters focused on these topics. Even Wikipedia is better. I suggest the Authors better describe this part.

Our intention was to be general and assumption-free, rather than naive. We had included citations for the interested reader of lattice defects and conformational disorder observed in molecular crystals.

We believe that figure 1 is simple and clear in showing what we are trying to convey: the hierarchical addition of different disorder components related to physical motions as is prominent in most crystals of biomacromolecules. Furthermore, the provided examples on SARS-CoV-2 Surface Glycoprotein and STEAP4 present the full complexity of disorder as present in protein structures.

(iii) Page 2, left column, bottom. The sentence “One convenient property of displacement parameters, U ,” mentions U , which has never been previously defined.

(3) *This point is the definition of U . We have added a clarification that we are defining U here.*

(iv) In the Discussion – page 8 right column – the Authors correctly write that “The output ECHT model is naturally dependent on the choice of levels, and how these levels are partitioned (e.g. choice of secondary structure). Therefore, groups must be chosen carefully and critical analysis of the model is essential.” This is absolutely true and crucial. Would it be possible to provide some strategies to “choose carefully” the groups to be analysed?

This is indeed a crucial point of the discussion and the reviewer is right to raise it. Complete and automated determination of the hierarchy is a direction of future work that we are actively pursuing.

(4) To address the currently raised point, we have expanded this section of the discussion to give practical advice, and pointed readers to the website for the program, where worked case studies will be uploaded.

(v) It might be interesting to compare ECHT results computed on the same crystal structure (same protein, same space group, same pH, etc.) determined at different resolutions. Actually, one can imagine that ECHT behaves and might be used differently at different resolution levels.

We agree with this and are indeed pursuing such analyses as part of future work. However, such an in-depth analysis is not currently feasible within this manuscript, which as reviewer two notes, already has an extensive supplementary component to ensure that the each of the comparative examples can be included. Additionally, the examples included here do allow for some of the comparison that the reviewer suggests: the structures of the SARS-CoV-2 main protease are determined at a range of resolutions and selected to show the effects of different B-factor refinements (iso, tls+iso, aniso); and the comparison of different temperatures addresses the scope of such an analysis, and points the way to future work.

Reviewer #2 (Remarks to the Author):

I consider this project and the results presented in the manuscript as extremely significant to the field of structural biology.

This is a kind of projects where a number of people may say “I was talking about this years ago...” but where such declaration means nothing by itself. The key result of this work is that the authors, first time for macromolecular structural studies, developed and practically realised a hierarchic model describing mobility of macromolecular structures. As the authors show by their examples, this both gives fundamental description of such mobility and suggests its role in molecular mechanisms (STEAP4, Fig. 7). The computer tools developed, being distributed within the major suites such as ccp4 and phenix, will allow the community of structural biologists to get a deeper understanding of structure-function relations.

We thank the reviewer for their very positive assessment of the work and its impact.

This key ‘ECHT method and program’ result should not hide another key issue, a long-time controversy if the TLS models indeed have some physical meaning or if the respective matrices are only formal ‘dummy’ parameters. Excellent examples presented in this work strongly support the former point.

(5) We agree with the reviewer, and, to their point, have included an additional comment in the discussion about the physical nature of TLS matrices.

I wonder if this method also may be used to validate atomic displacement parameters, ADPs. Their refinement seems to be an established procedure in X-ray crystallography but yet not so well in cryo-electron microscopy (cryo-EM). One can consider the models of SARS-Cov-2 surface glycoprotein as an example. In addition to the entries used in this work

(6vxx and 6vyb), PDB contains at least 7 other entries (6moj, 6wpt, 6xey, 6zb5, 6zp2, 7a97, 7kj2). In these models, the same residues have B-factors that vary from 20 to 400 Å². I believe that the ECHT approach can answer what is a physical significance of these ADPs, if any.

We agree that there are a variety of future directions for this work, including the validation of B-factors. One could certainly consider using the method as some quantification of “noise” in the atomic level as a measure of model quality, and we have added a note on this in the discussion (6).

Addressing the analysis the reviewer suggests, the global B-factor scale in cryo-EM models is poorly defined: From our understanding, the global B-factor scale is both a factor of the experimental disorder but also the (over-)sharpening of maps, leading to an “offset” in the average B-factor; we agree that an ECHT analysis would reveal the differences (if any) between these structures and once more an in-depth analysis would be very interesting, but once more likely outside the ability of this manuscript to address, and more suited to a more technical journal.

By all these reasons I expect that the scientific community will be happy to have an access, as soon as possible, to these results, both to the theory and the software.

An important feature of this method is that it does not necessary require high-resolution data and is applicable to structures solved at a resolution that varies in a quite large range. Also, the examples prove that the method is applicable to atomic models obtained by any of the two experimental techniques, crystallography or cryo-electron microscopy, naturally under condition that the refinement of ADPs has been done correctly.

The numerous Pymol-figures illustrate the decomposition of the atomic motion into the components of a different level. At my point of view, the authors could comment more on Figure 3a. It shows that, after a few cycles at a new level, the respective ADP contribution practically does not change anymore, and the source for a next level is always the “chain” one. I think this may be considered as an extra validation of the parameters, their robustness: one may interrupt the procedure at any level and be sure that the parameters at this and previous levels are correct or near correct (except “chain” that yet includes a contribution from non-analysed levels).

This is not necessarily true, though what the reviewer says is largely correct. Though indeed most of the “B-factor mass” comes from the chain level on subsequent cycles (due to the elastic net penalties, which “store” remaining B-factor mass in the chain level before “distribution” to lower levels), this does not mean that the “shapes” of the TLS groups do not change in subsequent cycles. The figure3a only shows the average over all of the groups in the level and therefore loses this detail. Additionally, the observation is not always true, and both the chain (or domain, if present) and secondary-structure levels may be the “sources” of B-factor-mass for “distribution” to lower levels.

The authors know well the literature in the field and efficiently use available algorithms, software and methods, with appropriate references to them. Their results, presented by

tables, figures and explained in text, support the conclusions and claims done by the authors. The method seems to be quite straightforward and therefore robust (the principal parameter is the weight for the elastic net penalties; am I right? Are there other parameters? – if yes, the authors need to clarify them).

The reviewer is correct – the only parameters (that the average user would adjust) are indeed the rate at which the elastic net is decayed, and the choice of levels, which are now discussed in the discussion at length (see change 4). How to use the program, and which parameters should be adjusted, are additionally discussed on the program website.

From my point of view, there are still several points where the work and the manuscript can be improved. As for other methods, there are questions: what to do 'in principle'? how to do this practically? how to validate the results? My main concern is the latter.

In the manuscript, the authors, experts in the field, validate their calculations by a careful structural inspection of the results. Since the tool certainly will be largely used, I may imagine a colleague who has no such large experience as the authors and who wished to use ECHT. How he / she can validate their results? How they may be sure that there is no over-ECHT-interpretation of the available ADP? I would suggest the authors of the manuscript to work more on the question of quantitative and objective validation of the ECHT results. How many data they have (Natoms in case of isotropic ADP or 6*Natoms in case of anisotropic ADPs, plus data from the elastic penalties, right?). How many parameters appear at each level (give the numbers for the examples considered in the manuscript)? It may be useful to remind a reader that the atomic coordinates are no longer adjustable parameters so one should not count them as unknowns.

The reviewer makes several good points. The problem with quantifying the complexity of the hierarchical disorder model, as pointed out in the text, is that most of the TLS groups are parametrically redundant. Therefore, the non-redundant parameter ratio is 1:1 for the input B-factors and ECHT model. (7) This is now clearly stated in the discussion. As such, the aim of the method is not to correct for errors in the input B-factors, but to apportion them to a particular length-scale. However, as we suspect the reviewer realises, there is the possibility for including additional restraints during model optimisation which would allow for "improvement" of the input B-factors, and would increase the data-parameter ratio, and this is yet another the direction of future work which requires a substantial amount of work.

Instead of the number of model parameters, the relevant quality criterion for ECHT models is the "model complexity" which is the total "B-factor mass" of the model components, that is, the sum over the average B-factors of each model component (e.g. atomic B-factor or TLS group). This model complexity is shown in the sum-of-amplitudes line in figure 3. It is clear that the terminology we used in the manuscript initially does not clarify this situation adequately, and so we have changed the language throughout the manuscript to discuss "model complexity" rather than the "number of model parameters" (8). We have also emphasised that the ECHT model will approximately reproduce the input B-factors, and that the non-redundant parameter ratio for the model is 1:1 (i.e. no reduction in model parameters) (7, again). As a matter of note, the numbers of model parameters are provided

in the output for the program, but we do not feel they are particularly meaningful, which is why they are not shown in the manuscript.

In light of these comments, the question of validating the ECHT model then becomes a question of two parts: when is the ECHT model correctly optimised; and when does the output from the ECHT model contain high-quality B-factors that can be safely interpreted. For the former, the output from the program outputs multiple graphs to help identify whether the optimisation proceeded smoothly, with appropriate descriptions in the automated HTML output and for all of the structures we have tested, it seems to “just work”, though there is certainly scope for more thorough validation, though this will require substantial additional work. For the second part, this becomes a question of the quality of the input B-factors, as it is these that arguably determines the quality of the ECHT model, and the interpretation of B-factors. The validation of B-factors is not, frankly, typically done in the field of structural biology. Instead, we rely on the implicit checking done by B-factor restraints in refinement. However, recent methods have been published for validation of B-factors that users could and should apply to the input and output models – these were initially cited in the introduction but we now cite them in the discussion to guide users in what they should do (9). To address the reviewer’s comment of what to do “in practise”, we now include a discussion on the use of automated re-refinement pipelines to obtain unbiased models (10).

By the way, the authors should correct the phrase between formula (9) and (10); the elastic penalties do not minimise the number of parameters but increase the number of ‘data’.

(8, again) We have changed the statement to discuss model complexity instead of the number of parameters.

Should the decomposition into levels stop as soon as the number of parameters becomes larger than the number of data, if this happens? Should it stop if the number of added parameters results only in a tiny modification of ADP, say, below 1% or is it indirectly included into the current conditions (section ‘Convergence’) ?

Yes, this is indirectly addressed by the convergence criteria, but we have found the convergence criteria generally provide an adequate cutoff. However, as we have discussed above, it is not the number of model parameters that is the cut-off, but rather then model complexity.

It would be great to give some statistical number that takes into account both the closeness of the UECHT to Uinput and the number of parameters.

(11) Yes – this is indeed a direction of future work, as discussed above. We have also added a note in the discussion that different ECHT decompositions can be compared using a normalised complexity, which is the model complexity divided by the number of atoms.

I have a few more small comments, mostly technical.

- As I learned, cryo-EM models typically have a larger disorder at their periphery due to uncertainties in the orientation of projections (“molecule level”). It looks like there is no such effect in 6hcy (Fig. 6). Could the authors comment more on this ?

This molecule is rather compact and is embedded in digitonin micelles. We suspect this is the reason that the translational component of disorder is dominant, with no quantifiable rotational disorder due to the compactness of the sample. We have added these points to the structural analysis (12).

- In section “Methods”, the authors use U for tensors (matrices in a given basis). What do they mean by $|U|$ in formula (5)? Its determinant? In formula (3), do they mean a (hidden) sum over the matrix elements? – these notations should be improved.

(13) The $|U|$ is the average of the isotropic B-factors for the generated U s. Our notation was unclear and we have changed it throughout to explicitly show the trace was intended.

- I expect that the coordinate basis of the L-matrix (section ‘TLS-level optimisation’) may vary with iterations; is it important for convergence ? Any comment?

We have not investigated the change in the matrices as a function of iteration but it could indeed be a very interesting study. We output the model after each iteration so such a study is very tractable. The perturbation of the matrices in the basis of the L-matrices is simply the obvious choice for choosing the basis matrices for the simplex method.

- It might be easier to follow the text if the principal section “Applications to Structural Analysis” has any kind of separation between the three examples.

(14) We tried to minimise the number of subheadings but we have introduced them now into the text – the editorial team may remove them as they wish.

- Is it correct that His41 is highlighted twice : by a transparent surface (hardly seen) in a right copy AND, conversely, by sticks in the left copy? Should this be clarified?

This was left in to allow the position of the HIS in both monomers to be seen, given that it is embedded within the monomer with ellipsoids shown, and that the copy in the other monomer is relevant when looking at the “back” of the protein in the supplementary figures. (15) We have amended the caption to state that both copies are shown.

- In (very) extended Supplementary Materials, figure captions 2, 4, 5, 10 refer to y-axis, and Fig 17 refers to x-axis while neither of these axes is indicated. Either the axes should be plotted or these phrases should be corrected.

We have added axes to each of the figures.

- I am unhappy with the presentation in Supplementary Figs. 7, 13, 14. I understand the idea of the authors, however giving ‘grey’ uninformative distributions and making each ‘black’ distribution in two (symmetric) copies does not give a better understanding but rather

confusing and these figures become in total too expanded. I suggest that the authors rethink and optimise these figures. This will allow them to increase the size of each individual informative 'black square.

We have removed the redundant plots.

My last query, which is below, may be ignored. However, since the authors act in the frame of 'pandemic' and since they already have the most of data in their hands, an answer on this query could be interesting, could complete the results of their project (unless the authors develop it as a small supporting opportune project) and be useful practically.

Site <https://en.wikipedia.org/wiki/EpiVacCorona> tells about one of the Covid-19 vaccines, and a Russian version of the same site, <https://ru.wikipedia.org/wiki/пиВакКорона>, shows the components of this vaccine (see also sites <https://patenton.ru/patent/RU2738081C1> and https://yandex.ru/patents/doc/RU2738081C1_20201207). The first component corresponds to the residues 454-477 of SARS-Cov-2 surface glycoprotein. These residues seem to be highly mobile and even missed in some of the nine models of this protein available in PDB (see above). Do the authors see anything particular in their results concerning this part of the protein? Maybe, some other models can give more information, such as 6m0j (crystal; resolution 2.45 Å), 6zp2 (resolution 3.10 Å) or 6zb5 (resolution 2.85 Å) ?

We would be very interested in performing such an analysis!

Alexandre Urzhumtsev

REVIEWER COMMENTS

Reviewer #2 (Remarks to the Author):

I think the authors tried their best and I am satisfied by their answers and modifications done in the manuscript.